# Development of UHPLC/Q-TOF Analysis Method to Screen Glycerin for Direct Detection of Process Contaminants 3-Monochloropropane-1,2-diol Esters (3-MCPDEs) and Glycidyl Esters (GEs)

**DOI:** 10.3390/molecules26092449

**Published:** 2021-04-22

**Authors:** Lauren Girard, Kithsiri Herath, Hernando Escobar, Renate Reimschuessel, Olgica Ceric, Hiranthi Jayasuriya

**Affiliations:** 1Center for Veterinary Medicine, Office of Research, Division of Residue Chemistry, Food and Drug Administration, 8401 Muirkirk Road, Laurel, MD 20708, USA; Lauren.Girard@fda.hhs.gov (L.G.); Kithsiri.Herath@fda.hhs.gov (K.H.); Hernando.EscobarLoaiza@fda.hhs.gov (H.E.); 2Center for Veterinary Medicine, Office of Research, Veterinary Laboratory Investigation & Response Network, Food and Drug Administration, 8401 Muirkirk Road, Laurel, MD 20708, USA; Renate.Reimschuessel@fda.hhs.gov (R.R.); Olgica.Ceric@fda.hhs.gov (O.C.)

**Keywords:** UHPLC/Q-TOF-MS analysis, glycerin, process contaminants, 3-monochloropropane-1,2-diol esters, glycidyl esters

## Abstract

The U.S. Food and Drug Administration’s (FDA′s) Center for Veterinary Medicine (CVM) has been investigating reports of pets becoming ill after consuming jerky pet treats since 2007. Renal failure accounted for 30% of reported cases. Jerky pet treats contain glycerin, which can be made from vegetable oil or as a byproduct of biodiesel production. Glycidyl esters (GEs) and 3-monochloropropanediol esters (3-MCPDEs) are food contaminants that can form in glycerin during the refining process. 3-MCPDEs and GEs pose food safety concerns, as they can release free 3-MCPD and glycidol in vivo. Evidence from studies in animals shows that 3-MCPDEs are potential toxins with kidneys as their main target. As renal failure accounted for 30% of reported pet illnesses after the consumption of jerky pet treats containing glycerin, there is a need to develop a screening method to detect 3-MCPDEs and GEs in glycerin. We describe the development of an ultra-high-pressure liquid chromatography/quadrupole time-of-flight (UHPLC/Q-TOF) method for screening glycerin for MCPDEs and GEs. Glycerin was extracted and directly analyzed without a solid-phase extraction procedure. An exact mass database, developed in-house, of MCPDEs and GEs formed with common fatty acids was used in the screening.

## 1. Introduction

In the late summer of 2007, the FDA became aware of reports of illness in dogs after consuming jerky pet treats (JPTs) [1,2]. For more than 10 years, the FDA dedicated extensive resources to the JPT investigation, including reviewing and investigating consumer complaint cases and collecting and testing JPT products (including products consumed by affected dogs) and animal diagnostic samples. Most of the cases (approximately 60%) involved gastrointestinal symptoms. Renal dysfunction or failure accounted for 30% of the reported cases, with a smaller subset reporting Fanconi syndrome, a dysfunction of the proximal renal tubules of the kidney in which glucose, amino acids, uric acid, phosphate, and bicarbonate are passed into the urine instead of being reabsorbed [3]. Fanconi syndrome is linked to exposure to certain toxins, medications, and infections in dogs as well as in people [4]. Exposure to toxins and drugs, including ethylene glycol (antifreeze), metals, chemotherapeutics, and expired tetracyclines and other antibiotics can damage the proximal renal tubule of the kidney [3,5]. Hooper commented that proximal renal tubular cells are exposed to the highest concentrations of renal-damaging substances, more so than distal tubular cells [6]. Damaged or dysfunctional proximal tubule cells can become “leaky”, allowing glucose, bicarbonate, amino acids, and other substances to cross into the urine by hindering normal resorptive functions.

JPTs are often made from dried chicken (or duck) meat or sweet potatoes. As of 1 September 2018, FDA’s Veterinary Laboratory Investigation and Response Network (Vet-LIRN) had collected and performed testing on more than 650 JPT samples related to more than 550 consumer complaints, as well as more than 400 retail samples of unopened product bags obtained from a store or shipment. Vet-LIRN did not subject every sample to the entire battery of testing due to limited resources and product availability. Sample testing was targeted based on the concerns with a particular product or brand and the symptoms displayed by the pet that consumed the product. The product-based testing plans targeted the main ingredients (chicken, duck, or sweet potatoes) and considered other information on the product labels, such as additional ingredients and/or their contaminants. For example, Vet-LIRN investigation revealed that some products contained glycerin (as high as 20%), although it was not listed on the label, and conducted additional testing for glycerin to follow up on this finding [2]. Glycerin is often added to JPTs to prevent excessive drying and improve the product’s texture.

Glycerin is also used to produce many other products including drugs, cosmetics, foods, tobacco products, sweeteners, and toiletries. As glycerin may be produced during biodiesel production and is subjected to a refining process, it is important to ensure that there are methods to detect potential toxic contaminants that could arise during manufacture.

Recently we published a UHPLC/Q-TOF method to screen crude glycerin for toxic phorbol ester contaminants [7]. Another group of compounds that could be contaminants in glycerin comprises 3-monochloropropane-1,2-diol (MCPD), its esters (MCPDEs), and glycidyl esters (GEs), which carry a reactive epoxide moiety. These compounds are formed during the industrial processing of oils in the presence of chloride ions, glycerin, and high temperatures [8]. In 2015, Vet-LIRN tested 74 JPT samples for MCPD and found that 31 samples tested positive at concentrations ranging from 0.027 to 0.352 ppm. The significance of this finding was not clear, although the review of the available scientific literature did not suggest that these MCPD levels in JPTs would cause illness in pets. 

Free MCPDs are present in the low mg/kg range in many foodstuffs such as acid-hydrolyzed vegetable protein, soy sauces, crackers, bread, toast and other bakery products, malt, meat products, and soups [9,10,11,12,13]. In most foodstuffs, only a small percentage of 3-MCPD is present as free 3-MCPD, while the majority is present as MCPDEs. Based on positive findings of MCPD in JPTs, Vet-LIRN investigation showed the need to evaluate JPTs and glycerin for their presence of MCPDEs; however, this type of testing is not available domestically. 

An extensive review about 3-MCPDE toxicity was published by the Joint FAO/WHO Expert Committee on Food Additives (JEFCA) in 2016 [14]. In 2018, the same regulatory body established a group total daily intake (TDI) of 2 μg/kg bw/day for 3-MCPD and its esters [15]. 3-MCPDEs are potentially toxic compounds with kidneys as their main target, according to reports from studies in animals. In Wistar rats or Swiss mice dosed with different MCPDEs, nephrotoxic degenerative and inflammatory changes in kidneys were evidenced by organ weight increase, glomerular lesions, tubulotoxicity (necrosis, tubular epithelial hyperplasia, and multifocal hypertrophy), cellular infiltration in interstitial spaces, and fibrosis [16,17]. Rat bioavailability studies show that 3-MCPDEs are completely hydrolyzed by enzymatic reactions in the gastrointestinal tract with release of 3-MCPD to be distributed to blood, organs, and tissues [18,19]. From short-term studies in rats and mice, the most affected organ after 3-MCPD induced toxicity was the kidney, with similar organ degenerative changes found for parent 3-MCPD esters [20]. A more recent report describes induced acute renal failure with nephrotoxic structural changes after oral administration of 3-MCPD in male albino rats in a period as short as 7 days [21].

Potential severe toxicological effects induced in animals with the presence of 3-MCPDEs have raised concerns about the level of exposure to these food contaminants. Strong evidence shows that glycidol and GEs are potential genotoxic carcinogen agents and probably carcinogenic to humans [22]. Studies on bioavailability have shown that GEs are hydrolyzed (de-esterified) during digestion, and the free glycidol is almost completely released [14]. The GEs are therefore treated like glycidol from a toxicological point of view. Due to the genotoxic potential of glycidol, it is not possible to derive any safe intake quantities for GEs.

Due to the potential renal toxicity of MCPDEs, which are derived from MCPD, and the carcinogenicity of glycidol derived from GEs, we focused our method development efforts to screen glycerin for MCPDEs and GEs.

## 2. Results and Discussion

Glycerin is the major byproduct of biodiesel production. Centrifugation or gravitational settling is used to separate biodiesel from glycerin after transesterification [23,24]. The lower glycerin phase consists of glycerol and other impurities such as methyl esters; water; alcohol; salts; and unreacted mono-, di-, and triglycerides. The crude glycerin is subjected to an expensive refining process that includes treatments such as bleaching, deodorizing, and ion exchange. Contaminants such as free fatty acids and their salts are removed to meet the USP standards to be used in food production. Some steps such as the deodorization step are carried out at elevated temperatures of more than 200 °C.

The main factors for the formation of MCPDEs and GEs are the presence of chloride ions; glycerin; and tri-, di-, or monoacylglycerides and high temperatures. As glycerin is subjected to these conditions during production and refining, it is possible that it is tainted with such process contaminants. 

Many edible oils also undergo a similar refining process to improve their quality. Evidence exists in the scientific literature to show the formation of MCPDEs and GEs during the industrial processing of oils [8,25,26,27]. These contaminants may be harmful to health and, therefore, undesirable in foods. There are many published studies to screen for these ester contaminants in edible oils [28,29,30], but to our knowledge, there are no methods published yet for glycerin. 

The determination of MCPDEs is complicated due to structural diversity. Considering the possible positional isomers of MCPD, the formation of about 100 different ester compounds is possible. However, because of the relative abundance of the fatty acids, only seven esters (lauric, myristic, palmitic, stearic, oleic, linoleic, and linolenic acids) are considered in food analysis [13]. For GEs, the number of ester structures is smaller, caused by its single hydroxyl group and lack of positional isomers.

LC method development was challenging due to the polarity range of the compounds. We tried to use ACN/water gradients spanning 5–100% ACN in 15 min with formic acid in both solvents. The GEs and the mono-MCPDEs, which are polar, eluted from the column, but the nonpolar di-MCPDEs did not elute from the column with this solvent gradient. We also tried other solvent gradients reported in the literature such as ACN/ MeOH/water as solvent A and acetone as solvent B ramping up to 60% acetone [31]. Even though the nonpolar di-MCPDEs eluted from the column with this solvent gradient, the peak shapes were broad. The IPA/8% aqueous methanol with formic acid step gradient used in the LC is a modification of a method reported in the literature for the analysis of GEs and mono- and di-MCPDEs as contaminants in edible oils [28]. LC parameters including mobile phase, column type, column temperature, flow rate, and gradient conditions were varied to establish the optimal liquid chromatography. The optimized step gradient allowed us to retain all three classes of compounds of differing polarities on the column as well as elute the most nonpolar di-MCPDEs within the gradient.

Even though we used six representative compounds of GEs and MCPDEs in our mixed standard, we analyzed many others in our study. The most polar 1-lauroyl-3-chloropropanediol and the most nonpolar 1,2-distearoyl-3-chloropropanediol eluted within our gradient. The LC retention times ranged from 2.3 to 10.3 min for the six compounds in our mixed standard using this gradient (Figure 1). Average responses in Table 1 indicated that the sodium adducts were much more dominant than the protonated adducts for all six compounds. This is true and consistent for all GEs and MCPDEs we analyzed during this study. This was an important observation that allowed us to screen for lower levels of MCPDEs by detecting unique ion clusters containing chlorine with sodium adducts. 

### 2.1. Calibration Curves

Calibration curves were generated for selected GEs and mono- and di-MCPDEs in both solvents, methanol and glycerin-matrix (Figure 2). All calibration standards were analyzed in triplicate over the concentration range of 10 to 400 ppb for MCPDEs and 2 to 100 ppb for GEs. A quadratic regression algorithm with no weighting was used to compare the standard curves in solvent versus matrix. The average correlation coefficients for all calibration curves were >0.997. We observed matrix enhancement for di- and mono-MCPD esters; however, glycidyl esters showed a matrix suppression effect. 

The calibration curves constructed for the mixed standard were used to determine the matrix effect. The calibration standards were used to validate the data analysis process.

Data analysis was performed with the Mass Hunter Qualitative Analysis software (B.06.00) using FbF (find by formula) algorithm to screen for compounds in the glycerin extracts using an in-house MCPDE and GE exact mass database as the formula source (Table 2). The in-house database was constructed by adding the exact masses of all possible esters of GEs and mono- and di-MCPDEs of lauric, myristic, palmitic, linolenic, linoleic, oleic, and stearic acids. The algorithm FbF was restricted to the *m/z* range of 50–1000 Da and completed within the timespan of the LC gradient. Singly charged ions and a minimum peak height of 10,000 ion counts were selected for FbF. The allowed adduct ions included H^+^ and Na^+^. FbF algorithm scores the database matches based on the similarity of each of the isotopic masses (Mass Match), isotope ratios (Abundance Match), isotope spacing (Spacing Match), and optionally, the retention time (RT Match). 

FbF algorithm found all six GEs and MCPDEs in the calibration standards, thereby validating the data analysis process (see Figure 1). Our method was capable of detecting GEs and di-MCPDEs down to 2 ng/mL and mono-MCPDEs down to 5 ng/mL concentrations. 

### 2.2. Analysis of Glycerin Samples

The refined and crude glycerin samples we obtained were diluted and subjected to the same screening procedure as the calibration standards. The screen against the database using FbF algorithm produced many false-positive hits for GEs that were eliminated by comparing the retention time and mass accuracy with standards. There were no hits for MCPDEs.

The data generated in the study were validated according to the HRMS guidance [32]. The mass accuracy was <5 ppm for all six compounds in the calibration standards. Each sample was analyzed in triplicate. The retention time shift was <0.2 min between the replicate injections. The detected levels were reproducible. The compound stability was monitored throughout the study by injecting the stock solutions of standards in methanol during the course of the study, and no instability was detected. 

## 3. Materials and Methods

### 3.1. Materials

#### 3.1.1. Instrumentation

Samples were analyzed using an Agilent 1290 Infinity UHPLC coupled with 6550 Q-TOF mass spectrometer (Agilent Technologies, Palo Alto, CA, USA) in positive-ion mode. 

#### 3.1.2. Chemicals

Glycidyl linolenate (CAS 51554-07-5), glycidyl linoleate (CAS 243085-63-3), 1-lauroyl-3-chloropropanediol (CAS 20542-96-5), 1-palmitoyl-3-chloropropanediol (CAS 30557-04-1), 2-oleoyl-3-chloropropanediol (CAS 915297-48-2), 1-palmitoyl-2-stearoyl-3-chloropropanediol (CAS 1185060-41-6), 1-oleoyl-2-stearoyl-3-chloropropanediol (CAS 1336935-05-7), and 1,2-distearoyl-3-chloropropanediol (CAS 72468-92-9) were purchased from Toronto Research Chemicals (Toronto, ON, Canada).

#### 3.1.3. Glycerin Samples

Sixteen food-grade glycerin samples and four crude-grade glycerin samples were obtained from various sources. A food-grade glycerin sample obtained from Procter & Gamble (Cincinnati, OH, USA) was used for constructing matrix calibration curves for the mixed standard.

### 3.2. Preparation of Stock Solutions and Working Standards

Primary stock solutions (1 mg/mL each) were prepared separately by dissolving the ester standard in methanol. There are two classes of MCPDEs, namely mono- and diesters. Two GEs (glycidyl linoleate and glycidyl linolenate), two MCPD monoesters (1-palmitoyl-3-chloropropanediol and 2-oleoyl-3-chloropropanediol), and two MCPD diesters (1-oleoyl-2-stearoyl-3-chloropropanediol and 1-palmitoyl-2-stearoyl-3-chloropropanediol) were selected to represent the two classes of MCPDEs and GEs. Working standard solutions were prepared at 50, 100, 500, and 1,000 µg/mL by diluting appropriate volumes of the primary stock solution with MeOH. All solutions were stored in glass vials at 4 °C or below.

### 3.3. Preparation of a Calibration Curve of the Mixed Standard in Glycerin and Methanol

Appropriate amounts of working standards of MCPDEs and GEs were spiked (added) into methanol and glycerin matrix to produce the following concentrations: 2, 5, 10, 25, 50, 125, 250, and 400 ng/glycerin. Mixed standard in glycerin was prepared using 1 g food grade glycerin (Procter and Gamble) in 2 mL methanol, spiked with appropriate volumes of working standards. 

Calibration curves were constructed in both methanol and glycerin matrix.

### 3.4. Glycerin Sample Preparation for Screening for MCPDEs and GEs

Sixteen food-grade glycerin samples and four crude-grade glycerin samples (approximately 1 g each) were prepared separately in 4 mL glass amber vials. Each 1 g glycerin was weighed directly into the vial, and 500 µL of methanol was pipetted into each vial. Vials were capped and vortexed, and methanol was added dropwise to bring the total volume of each sample up to the 2 mL mark. Methanol was used as the solvent blank. Triplicate injections were performed for each sample. 

### 3.5. UHPLC Analysis

We analyzed the diluted spiked glycerin without solid-phase extraction step using the QTOF6500 HRMS mass spectrometer coupled to the UHPLC system. A liquid chromatography system 1290 (Agilent Technologies, Santa Clara, CA, USA) with a BEHC18 column (2.0 mm × 100 mm) with 1.8 μm particles (Waters, Milford, MA, USA) at 30 °C was used for HPLC separation. Mobile phase A consisted of 92:8 methanol/water with 0.05% formic acid; mobile phase B consisted of 98:2 isopropanol/water with 0.05% formic acid. The flow rate was 300 µL/min, and the gradient used was as follows: from 0–3 min, mobile phase B 0–30%; from 3–10 min, mobile phase B 30–70%, followed by holding at 70% B for 5 min and returning to 0% B in 2 min. The first 1 min of the LC flow was diverted to the waste to prevent the glycerin matrix from entering the mass spectrometer.

### 3.6. HRMS Analysis

The Agilent 6550 QTOF instrument (Agilent Technologies, Santa Clara, CA, USA) is equipped with an Agilent Jet Stream Technology Dual Spray ESI source and an iFunnel. The instrument was calibrated in the extended dynamic range (2 GHz, High Res Mode) and lower mass range (*m/z* < 1700) in the positive ion mode. Data were collected in centroid format. Reference masses at *m/z* 121.0509 and *m/z* 922.0089 were continually introduced via a second sprayer for accurate mass calibration. The reference ions used were purine (C_5_H_4_N_4_) at *m/z* 121.0509 and HP-921 (hexakis-(*1H*,*1H*,*3H*-tetrafluoro-pentoxy)phosphazene (C_18_H_18_O_6_N_3_P_3_F_24_)) at *m/z* 922.0089 for positive mode. The source conditions were as follows: sheath gas flow, 11 L min^−1^; sheath gas temperature, 350 °C; nebulizer pressure, 40 psi; drying gas temperature, 150 °C; drying gas flow, 15 L min^−1^; nozzle voltage, 380 V; fragmentor voltage, 360 V; capillary voltage, 3500 V. Nitrogen was used as source gas and as collision gas. Full scan MS were acquired over the mass range *m*/*z* 100–1000 at a scan speed of 2 scans/s. Agilent Mass Hunter workstation software version B.05.00 was used for data acquisition, and B.06.00 qualitative analysis was used for processing. Mass calibration was performed prior to analysis.

## 4. Conclusions

We accomplished our goal in this study by successfully developing a UHPLC/Q-TOF-based screening method to detect a variety of GEs and MCPDEs in glycerin samples (both crude and refined grades). It is a dilute-and-shoot method with minimal sample preparation that can be adapted to other matrices. The screening method by FbF against our in-house library produced many false positives for GEs, which were eliminated by comparing the retention times with standards. There were no hits for MCPDEs in the limited glycerin samples we analyzed. 

## Figures and Tables

**Figure 1 molecules-26-02449-f001:**
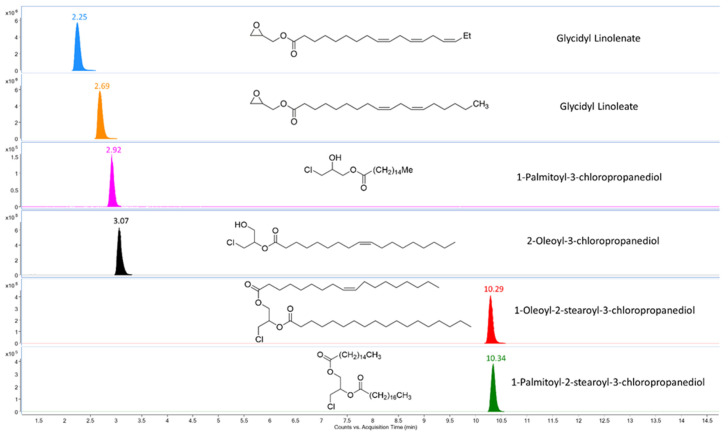
Extracted ion chromatograms for the mixed standard of glycidyl esters (GEs) and 3-monochloropropanediol esters (MCPDEs) in the glycerin matrix.

**Figure 2 molecules-26-02449-f002:**
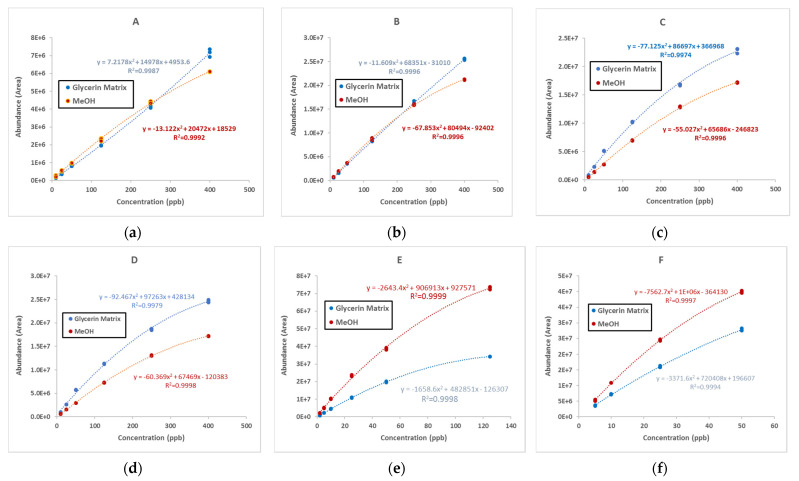
Calibration curves prepared in glycerin matrix and methanol for (**a**) palmitoyl-chloropropanediol, (**b**) oleyl-chloropropanediol, (**c**) 1-oleoyl-2-stearoyl-2-chloropropanediol, (**d**) 1-palmitoyl-2-stearoyl-3-chloropropanediol, (**e**) glycidyl linoleate, and (**f**) glycidyl linolenate.

**Table 1 molecules-26-02449-t001:** Average response for protonated and sodium adducts of GEs and MCPDEs in the mixed standard.

Compound	Exact Mass	(M+H)	(M+Na)
Glycidyl linolenate	334.2508	2,209,671	34,331,876
Glycidyl linoleate	336.2664	3,308,840	45,741,444
1-Palmitoyl-3-chloropropanediol	348.2431	0	1,517,938
2-Oleoyl-3-chloropropanediol	374.2588	116,159	6,265,336
1-Oleoyl-2-stearoyl-3-chloropropanediol	640.5197	0	7,484,321
1-Palmitoyl-2-stearoyl-3-chloropropanediol	614.5041	0	8,217,162

**Table 2 molecules-26-02449-t002:** Database of MCPDEs and GEs.

Formula	Exact Mass	Compound Name
**MCPD monoesters**		
C17H33ClO3	320.2118	Myristoyl-chloropropanediol
C15H29ClO3	292.1805	Lauroyl-chloropropanediol
C19H37ClO3	348.2431	Palmitoyl-chloropropanediol
C21H41ClO3	376.2744	Stearoyl-chloropropanediol
C21H39ClO3	374.2588	Oleyl-chloropropanediol
C21H37ClO3	372.2431	Linoleoyl-chloropropanediol
C21H35ClO3	370.2275	Linolenoyl-chloropropanediol
**MCPD diesters**		
C35H67ClO4	586.4728	Di-palmitoyl-chloropropanediol
C27H51ClO4	474.3476	Di-lauroyl-chloropropanediol
C29H55ClO4	502.3789	Lauroyl-myristoyl-chloropropanediol
C33H57ClO4	552.3945	Lauroyl-linolenoyl-chloropropanediol
C33H59ClO4	554.4102	Lauroyl-linoleoyl-chloropropanediol
C39H67ClO4	634.4728	Di-linoleoyl-chloropropanediol
C39H63ClO4	630.4415	Di-linolenoyl-chloropropanediol
C35H61ClO4	580.4285	Linolenoyl-myristoyl-chloropropanediol
C31H59ClO4	530.4102	Di-myristoyl-chloropropanediol
C31H59ClO4	530.4102	Lauroyl-Palmitoyl-chloropropanediol
C33H61ClO4	556.4258	Lauroyl-Oleoyl-chloropropanediol
C39H65ClO4	632.4571	Linoleoyl-Linolenoyl-chloropropanediol
C35H63ClO4	582.4415	Myristoyl-Linoleoyl-chloropropanediol
C33H63ClO4	558.4415	Myristoyl-palmitoyl-chloropropanediol
C33H63ClO4	558.4415	Lauroyl-stearoyl-chloropropanediol
C35H65ClO4	584.4571	Myristoyl-oleoyl-chloropropanediol
C39H71ClO4	638.5041	Stearoyl-linoleoyl-chloropropanediol
C39H73ClO4	640.5197	Oleoyl-stearoyl-chloropropanediol
C39H71ClO4	638.5041	Linoleoyl-stearoyl-chloropropanediol
C37H71ClO4	614.5041	Palmitoyl-stearoyl-chloropropanediol
C37H67ClO4	610.4728	Palmitoyl-linoleoyl-chloropropanediol
C37H65ClO4	608.4571	Palmitoyl-linolenoyl-chloropropanediol
C35H67ClO4	586.4728	Myristoyl-stearoyl-chloropropanediol
C39H71ClO4	638.5041	Di-oleoyl-chloropropanediol
C37H69ClO4	612.4884	Oleoyl-palmitoyl-chloropropanediol
C39H75ClO4	642.5354	Di-stearoyll-chloropropanediol
C39H67ClO4	634.4728	Oleoyl-linolenoyl-chloropropanediol
C39H69ClO4	636.4884	Oleoyl-linoleoyl-chloropropanediol
C37H69ClO4	612.4884	Oleyl-palmitoyl-chloropropanediol
**Glycidyl esters (GEs)**		
C15H28O3	256.2038	Glycidyl laurate
C17H32O3	284.2351	Glycidyl myristate
C21H40O3	340.2977	Glycidyl stearate
C19H36O3	312.2664	Glycidyl palmitate
C21H38O3	338.2821	Glycidyl oleate
C21H36O3	336.2664	Glycidyl linoleate
C21H34O3	334.2508	Glycidyl linoleate

## Data Availability

Not applicable.

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
