# Peer review of "Development of UHPLC/Q-TOF Analysis Method to Screen Glycerin for Direct Detection of Process Contaminants 3-Monochloropropane-1,2-diol Esters (3-MCPDEs) and Glycidyl Esters (GEs)"

_molecules, 2021, doi:10.3390/molecules26092449_

Round 1

Reviewer 1 Report

The manuscript described the development of the UHPLC/Q-TOF method for screening glycerin for MCPDEs and GEs. The manuscript could be accepted after minor revise.

  1. In first three paragraphs of the introduction part, relative references should be added.
  2. In page 3-5. Authors put too much emphasis on background introduction. More description of results (Figure1-2 and Table 1) and discussion of the results should be added, especially of Figure 1 and Table 1.
  3. Check the details and format of references. For example, Ref. 10, EFSA-CONTAM should be added to the authors list. The underline was added to doi of Ref. 20 but not to other References, please unify according to the requirements of journal.

Reviewer 2 Report

The research article "Development of UHPLC/Q-TOF Analysis Method to Screen Glycerin for Direct Detection of Process Contaminants 3-monochloropropane-1, 2-diol esters (3-MCPDEs) and Glycidyl Esters (GEs)" deal with the development of a UHPLC/Q-TOF method for the screening of Glycidyl esters (GEs) and 3-monochlo-ropropanediol esters (3-MCPDEs) in glycerin.

After the evaluation of your manuscript, I regret to inform you that in my opinion this manuscript is not suitable for publication in molecules. Despite a very-well written introduction and a good focus on the aim of the study, several pivotal parts are missing:

  • No description regarding the optimization of the UHPLC condition is provided except the following paragraph "LC method development was challenging due to the polarity range of the com-pounds. We tried to use ACN/ Water gradients spanning 5-100% ACN in 15 min with TFA in both solvents, but the nonpolar MCPD diesters could not elute from the column. The IPA/ 8% aqueous methanol with TFA step gradient used in the LC allowed us to retain compounds of differing polarities on the column." Why have been these solvents used? Were these conditions optimized using an experimental design or an optmization method?
  • Validation is very poor: repeteability, detection/quantification limit and method trueness are not provided
  • No data regarding real sample analysis have been provided (quantitation of the analytes or qualitative analysis)
  • Despite the database contains more than 40 analytes only six standards have been analyzed for quantitation
  • No conclusion or discussion is present

Reviewer 3 Report

The manuscript entitled “Development of UHPLC/Q-TOF Analysis Method to Screen Glycerin for Direct Detection of Process Contaminants 3-mono-chloropropane-1, 2-diol esters (3-MCPDEs) and Glycidyl Esters (GEs)” is very interesting and could have a great impact on the analysis of food contaminants. The paper is very well structured, and all experiments are conducted in a good manner. There are only a few changes that should be changed before publication.

Please check through the text and add space between value and unit (Page 3, section 2; Page 6 sub-sections 3.2., 3.4., 3.5., 3.6.). Also, in sub-section 3.5. change symbol after 1.8 value with the appropriate Greek symbol. There is some inconsistency in References style so please check and correct.  

Round 2

Reviewer 2 Report

After the evaluation of the revised version of the manuscript Development of UHPLC/Q-TOF Analysis Method to Screen Glycerin for Direct Detection of Process Contaminants 3-monochloropropane-1, 2-diol esters (3-MCPDEs) and Glycidyl Esters (GEs) I find that the authors provided sufficient revision in the article to be published in Molecules. 

I have just one point to address: in paragraph 4, page 3 (lines 144-148) details regarding the choice of the eluent phase have been added. However, if possible please provide more information for the optimized parameters, i.e. types of columns, ranges of gradients and composition of the mobile phase, temperatures,... Moreover if no experimental design/optimization using a statistical model/method have been reported please replace "optimized" with proper/described/...

However I find that the edits provided are sufficient for publication

best regards